# Lacking Immunotherapy Biomarkers for Biliary Tract Cancer: A Comprehensive Systematic Literature Review and Meta-Analysis

**DOI:** 10.3390/cells12162098

**Published:** 2023-08-19

**Authors:** Giorgio Frega, Fernando P. Cossio, Jesus M. Banales, Vincenzo Cardinale, Rocio I. R. Macias, Chiara Braconi, Angela Lamarca

**Affiliations:** 1Osteoncology, Soft Tissue and Bone Sarcomas, Innovative Therapy Unit, IRCCS Istituto Ortopedico Rizzoli, 40136 Bologna, Italy; giorgio.frega@ior.it; 2Department of Organic Chemistry I, Center of Innovation in Advanced Chemistry (ORFEO-CINQA), University of the Basque Country/Euskal Herriko Unibertsitatea (UPV/EHU), Donostia International Physics Center (DIPC), 48940 Donostia-San Sebastian, Spain; fp.cossio@ehu.es; 3Department of Liver and Gastrointestinal Diseases, Biodonostia Health Research Institute—Donostia University Hospital, University of the Basque Country (UPV/EHU), Ikerbasque, 48940 San Sebastian, Spain; jesus.banales@biodonostia.org; 4National Institute for the Study of Liver and Gastrointestinal Diseases (CIBERehd), Instituto de Salud Carlos III, 28029 Madrid, Spain; rociorm@usal.es; 5Department of Biochemistry and Genetics, School of Sciences, University of Navarra, 31009 Pamplona, Spain; 6Department of Medico-Surgical Sciences and Biotechnologies, Sapienza University of Rome, 00185 Rome, Italy; vincenzo.cardinale@uniroma1.it; 7Experimental Hepatology and Drug Targeting (HEVEPHARM), IBSAL, University of Salamanca, 37007 Salamanca, Spain; 8School of Cancer Sciences, University of Glasgow, Glasgow G12 8QQ, UK; chiara.braconi@glasgow.ac.uk; 9Beatson West of Scotland Cancer Centre, Glasgow G12 0YN, UK; 10Department of Oncology—OncoHealth Institute, Instituto de Investigación Sanitaria de la Fundación Jiménez Díaz, Fundación Jiménez Díaz University Hospital, 28040 Madrid, Spain; 11Department of Medical Oncology, The Christie NHS Foundation, Manchester, Division of Cancer Sciences, University of Manchester, Manchester M13 9PL, UK

**Keywords:** biliary cancer, immunotherapy, predictive biomarkers

## Abstract

Background: Immunotherapy has recently been incorporated into the spectrum of biliary tract cancer (BTC) treatment. The identification of predictive response biomarkers is essential in order to identify those patients who may benefit most from this novel treatment option. Here, we propose a systematic literature review and a meta-analysis of PD-1, PD-L1, and other immune-related biomarker expression levels in patients with BTC. Methods: Prisma guidelines were followed for this systematic review and meta-analysis. Eligible studies were searched on PubMed. Studies published between 2017 and 2022, reporting data on PD-1/PD-L1 expression and other immune-related biomarkers in patients with BTC, were considered eligible. Results: A total of 61 eligible studies were identified. Despite the great heterogeneity between 39 studies reporting data on PD-L1 expression, we found a mean PD-L1 expression percentage (by choosing the lowest cut-off per study) of 25.6% (95% CI 21.0 to 30.3) in BTCs. The mean expression percentages of PD-L1 were 27.3%, 21.3%, and 27.4% in intrahepatic cholangiocarcinomas (iCCAs—15 studies), perihilar–distal CCAs (p/dCCAs—7 studies), and gallbladder cancer (GBC—5 studies), respectively. Furthermore, 4.6% (95% CI 2.38 to 6.97) and 2.5% (95% CI 1.75 to 3.34) of BTCs could be classified as TMB-H and MSI/MMRd tumors, respectively. Conclusion: From our analysis, PD-L1 expression was found to occur approximately in 26% of BTC patients, with minimal differences based on anatomical location. TMB-H and MSI molecular phenotypes occurred less frequently. We still lack a reliable biomarker, especially in patients with mismatch-proficient tumors, and we must need to make an effort to conceive new prospective biomarker discovery studies.

## 1. Introduction

Biliary tract cancers (BTCs) refer to a variety of rare and aggressive cancers that arise from the biliary epithelial tree both within the liver (iCCA) and outside the liver, with the latter comprising perihilar (pCCA) and distal (dCCA) cholangiocarcinoma. Gallbladder cancer (GBC) and the ampulla of Vater cancer (AVC) are also encompassed under BTC. Among primary liver cancers, iCCA is the second most common type, after hepatocellular carcinoma [1]. There are significant geographical discrepancies in the incidence of iCCA and p/dCCA which reflect relevant differences in host genetics and local risk factors [2]. In Western nations, the incidence of iCCA is on the rise [3]. Overall, BTCs lack effective treatments due to predominant diagnoses at advanced stages. The only potential curative option is surgery when feasible. Locally advanced and metastatic patients face a poor prognosis. Personalized treatments, e.g., target therapies, are only available for limited subgroups and their impact is far more limited than in other neoplasms.

In 2022, the FDA and EMA approved immunotherapy using an anti-programmed death ligand 1 (PD-L1) antibody based on the results of the TOPAZ-1 trial [4]. This trial demonstrated an improvement in overall survival (OS), with a hazard ratio (HR) of 0.80 (95% CI, 0.66–0.97; *p* = 0.021) when durvalumab was added to the first-line standard of care (gemcitabine plus cisplatin). Notably, an estimated 24.9% of patients were still alive at 24 months, compared to 10.4% in the control group [5]. In addition, results from the phase III KEYNOTE-966 were recently published, also showing an advantage in terms of OS in favor of adding pembrolizumab to cisplatin and gemcitabine over the placebo (HR: 0.83; 95% CI, 0.72–0.95; one-sided *p* = 0.0034) [6].

The identification of reliable predictive biomarkers is crucial for selecting patients who will benefit most from both currently approved and investigational immune-based approaches [7]. Even in cancers classified as responsive to immunotherapy, only a proportion of patients experience a substantial and long-lasting benefit [8].

To date, several molecular and genetic tumor features have been proposed in order to predict therapeutic responses, sometimes with discordant results. Moreover, even the composition of the patient’s microbiota may have an impact on cancer immunotherapy [9]. PD-L1 expression can partially predict the magnitude of benefits that derive from immunotherapy, but a negative expression is not always indicative of a lack of a response [10]. Furthermore, there is significant heterogeneity regarding the cut-off scores and PD-L1 immunostaining data. Four different PD-L1 antibodies (22C3, 28–8, SP263, and SP142) were approved by the FDA [11], and notably the concordance between different assays is moderate [12]. Some studies and metanalyses in patients with BTC showed that programmed cell death protein 1 (PD-1) and PD-L1 expression is associated with poor prognosis and advanced disease [12,13,14,15]. Other reports found no statistical correlation or even a favorable prognostic role for PD-L1 expression in immune cells [16]. In addition, none of the studies supporting the use of immune checkpoint inhibitors (ICIs) in BTC mentioned above (TOPAZ-1 and KEYNOTE-966) have been able to identify any link between the PD-L1 status and the treatment-derived benefit. Thus, biomarkers for patient selection are urgently required.

Tumor mutation burden (TMB) represents the total number of somatic mutations per megabase, including nonsynonymous mutations, insertions, and deletions. A high TMB (TMB-H) correlates with the response to ICIs in several types of cancer [17]. This is probably due to the formation of a large amount of immune neoantigens, and ultimately in increased tumor immunogenicity [18,19]. The optimal cut-off for TMB-H definition has been a matter of debate. The most commonly used cut-off value is ≥10 Mut/Mb based on the FDA agnostic approval of anti-PD-1 therapy. Moreover, there is significant heterogeneity in the cut-off values used in different retrospective series. Despite the fact that its predictive value may vary across different cancer types [20,21], its predictive role at a cut-off value of ≥10 Mut/Mb has been recently validated in real-world data in a wide number of solid tumors [22].

Deficiency in mismatch repair (MMR) is typically identified through immunohistochemical staining. A lack of expression in one of the MMR proteins (MLH1, MSH2, MSH6, and PMS2) has been reported to occur in less than 10% of samples and often correlates with high microsatellite instability (MSI) status after DNA testing (with approximately 70% concordance) [23]. The MSI phenotype has been reported in <5% of BTCs [24] with a slightly higher incidence in iCCA than in p/dCCA and GBC [25]. Both of these alterations have been correlated with a cancer immunotherapy response in basket trials in non-colorectal cancers, including BTC. The most representative example is the KEYNOTE-158 clinical trial, which included a total of 22 BTC patients with MMR deficiency or MSI high tumors and achieved a complete and partial radiological response rate of 13.6% and 27.3%, respectively [26].

Here, we present a systematic review of the existing literature on immunotherapy biomarkers in BTCs, particularly to define their prevalence in BTC and in each anatomical subtype (iCCA, p/dCCA, and GBC).

## 2. Methods

### 2.1. Objectives

The primary objective of our analysis was to identify the prevalence of immune-related biomarkers in BTCs. The following biomarkers were included: PD-L1 expression; tumor mutation burden; DNA MMR; MSI; tumor-infiltrating lymphocytes (TILs); IFN-γ signaling pathways; and neoantigen load.

### 2.2. Search Strategies

We performed a search on PubMed. We employed the following search strings: (“PD-1” OR “PD-L1” OR “PD1” OR “PDL1” OR “Programmed Cell Death 1 Receptor”[Mesh]) AND (“cholangiocarcinom*” OR “Klatskin Tumor” OR “Gallbladder cancer” OR “Cholangiocarcinoma”[Mesh] OR “Gallbladder Neoplasms”[Mesh] OR “Klatskin Tumor”[Mesh]); (“Tumor mutation burden” OR “mismatch repair” OR “MMR” OR “microsatellite instability” OR “MSI” OR “neoantigen load” OR “DNA Mismatch Repair”[Mesh] OR “Microsatellite Instability”[Mesh]) AND (“cholangiocarcinom*” OR “Klatskin Tumor” OR “Gallbladder cancer” OR “Cholangiocarcinoma”[Mesh] OR “Gallbladder Neoplasms”[Mesh] OR “Klatskin Tumor”[Mesh]); (“Tumor infiltrating lymphocytes” OR “TIL” OR “IFN-γ” OR “Lymphocytes, Tumor-Infiltrating”[Mesh] OR “Interferon-gamma”[Mesh]) AND (“cholangiocarcinom*” OR “Klatskin Tumor” OR “Gallbladder cancer” OR “Cholangiocarcinoma”[Mesh] OR “Gallbladder Neoplasms”[Mesh] OR “Klatskin Tumor”[Mesh]). 

### 2.3. Study Eligibility

Inclusion criteria were as follow: studies that included patients with iCCA, p/dCCA, GBC, or AVC; retrospective/prospective design; studies reporting data on immune-related biomarkers of interest identified for this review, including prevalence and/or predictive/prognostic impacts (e.g., tumor response or progression-free survival (PFS) and OS); studies published on PubMed from January 2017 to January 2022. Exclusion criteria were as follows: reviews and opinion articles; studies with fewer than 10 patients; studies published in a language other than English; studies that not reported data on at least one of the following immune biomarkers: PD-L1 expression, TMB, DNA MMR, MSI, TILs, IFN-γ signaling pathways, and neoantigen load; and studies on others tumor types without any distinction for BTC. 

If there were duplicates (studies including the same population), the publication that included the largest number of patients was selected, and the other was excluded. If a multicenter study included a single-center series that was published separately, the multicenter study was selected and the single-center series was discarded. If a study reported data on specific subgroups of patients (e.g., EBV-positive patients, combined hepatocellular cholangiocarcinoma, etc.), those data were excluded when possible. If a study investigated the biomarkers of interest only in a subgroup of included patients, the percentage was calculated on that numerosity. For each biomarker, we included data for the lowest cut-offs available. To be more specific, for PD-L1, we considered positive cases with the lowest cut-off considered by the single study. Furthermore, if reported, we extracted the data on the expression on tumor cells. For TMB, we considered TMB-H tumors as established by the cut-off defined in each study (if the cut-off was not reported, the data were extrapolated when possible, using a cut-off of 20 mut/Mb). Finally, for dMMR/MSI-H analysis, we considered both cases that were reported as dMMR by immunohistochemistry (IHC) and/or MSI-H by PCR or gene sequencing to be positive. This scoping review was conducted following PRISMA guidelines.

### 2.4. Quality Assessment and Evaluation of the Risk of Bias

The reliability of the studies for our objective was determined by assessing the risk of bias. The study was considered to be of good (G) reliability if no more than two high-risk-of-bias features were detected; of medium (M) reliability if two to four high risk-of-bias features were detected; and of poor (P) reliability if more than four high risk-of-bias features were detected. The indicators for the assessment of risk of bias are presented in Table 1.

### 2.5. Statistical Analysis 

Descriptive statistical analysis was performed using Excel. Graphs and statistical analyses of biomarkers (e.g., calculation of the weighted mean value, taking into account the number of patients per study) were performed using Excel and R studio (Version 1.2.504; ggplot2, dplyr, hrbrthemes, viridis, and gapminder packages).

## 3. Results

### Selection of the Studies

After combining the results of each search string, 308 items were retrieved. We excluded 68 duplicates, resulting in 240 results. After assessing the titles or abstracts, 107 items were considered for full text evaluation. Twenty-eight manuscripts were excluded after full-text evaluation. Then, 18 of the remaining 79 were excluded mainly because they focused on different biomarkers. Eventually, 61 studies were analyzed. Figure 1 shows a flowchart of the study. 

This study included a total of 61 eligible studies, most of which were retrospective (90%) and unicentric. The reliability of the studies for our objective was limited, with most being of medium reliability (65.6%), mainly due to the retrospective aspect of design (90% of eligible studies), the monocentric aspect of the study (82% of eligible studies), possible bias in the selection of patients (11% of eligible studies), and no central or blinded review of the biomarkers (46% of eligible studies). In terms of the primary tumor type, most included either iCCA (34.4%) or a mixture of primary BTC tumors (39.3%). p/dCCA and GBC were less represented (11.5% and 14.8% of studies, respectively). In our review, we included 39, 13, and 27 studies reporting data on PD-L1 expression, TMB, and MSI/dMMR, respectively. No relevant studies to allow pooled analysis for TILs, IFN-γ signalling pathways, and neoantigen load pathways were identified. The details studied for each subgroup are summarized in the Appendix A.

The characteristics of the included studies are outlined in Table 2.

A total of 13,490 patients were included in the 61 considered studies, whereby 6714 patients had iCCA, 2083 had p/dCCA (including 448 Klatskin tumors and 779 distal p/dCCA), 2506 had GBC, 37 had AVC carcinoma, and 923 had other or unspecified histology. Data from 12,870 patients were considered (some patients were excluded mainly due to mixed histology such as hepatobiliary carcinoma or restricted molecular subgroups). The mean number of patients per study was 221 (95% CI 151 to 292). Furthermore, 5972 out of 12,638 patients were female (47.2%). The mean age (51 studies) was 62.7 years (95% CI 61.4 to 64.1). The disease stage was specified for 2568 patients, 1289 of whom had stage I-II disease, while 1279 had stage III-IV disease. In total, 4131 patients were resected. The patient characteristics are summarized in Table 3.

## 4. PD-L1

Thirty-nine studies with PD-L1 expression data were included. We acknowledge that a great heterogeneity of PD-L1 expression data emerged due to the cut-off points chosen by each study, the dilution and type of antibody used, and the reported data on expression (PD-L1 expression on tumor cells, CPS—“combined positive score”, and TPS—“tumor proportion score”). We decided to exploit the lowest level of expression per study, as well as to privilege data on PD-L1 expressions on tumor cells when provided, instead of combined expression scores. To be more specific, 11/39 studies used 1% of tumor cells as the cut-off; 10/39 used ≥5% of tumor cells; 1/39 used ≥2% of tumor cells; 1/39 used ≥25% of tumor cells; 2/39 used CPS ≥ 1%; 2/39 used TPS ≥ 1%; 5/39 used combined scores; and 7/39 used unspecified scores. 

The mean percentage of PD-L1-positive tumors among the studies was 25.6% (95% CI 21.0 to 30.3). The median percentage of PD-L1-positive tumors among the studies was 23%. The weighted mean percentage of PD-L1-positive tumors was 21.7%.

The mean percentages of PD-L1 expression were 27.3%, 21.3%, and 27.4% in iCCAs (15 studies), p/dCCAs (7 studies), and GBC (5 studies), respectively.

The percentage of PD-L1-positive tumors in each study is shown in Appendix A and Figure 2.

## 5. TMB

Thirteen studies with data on TMB were included. The mean number of patients per study was 484. We acknowledge that great heterogeneity in TMB data due to the different cut-off points used in the studies. To be more specific, six, three, and four studies used 19.5–20 mut/Mb, 17 mut/Mb, and 9.3–12.5 mut/Mb as cut-off points, respectively. The mean mut/Mb was reported by 10 studies.

The mean and median TMB values were 4.9 mut/Mb and 4.7 mut/Mb, respectively.

The mean TMB-H percentage per patient in all 13 studies was 4.6% (95% CI 2.38 to 6.97). The median percentage of TMB-H patient was 3.8%. The weighted mean percentage of TMB-H patients in the 13 studies was 3.6%. The percentage of TMB-H tumors in each study is shown in Appendix A and Figure 3.

## 6. MSI/dMMR

We identified 27 studies that investigated MSI/dMMR status. The mean number of patients per study was 338. We recognize that there was significant heterogeneity in MSI/dMMR due to the specific methodologies (IHC/NGS/PCR) proposed by various authors.

The mean percentage of patients defined as dMMR/MSI among the studies was 2.5% (95% CI 1.75 to 3.34). The median percentage of dMMR/MSI patients was 2.1%. The weighted median was 2.07%. The percentage of dMMR/MSI tumors in each study is presented in Appendix A and Figure 4.

## 7. Discussion

Data on immunotherapy biomarkers are yet to be conclusive for BTC. Our systematic review and pooled analysis confirm that MMR deficiency/MSI and high TMB in BTC, which likely represent the most reliable predictive biomarkers for immunotherapy, are rare (4.6% TMB-H; 2.5% MSI/MMRd). Despite the expression of PD-L1 being more frequent (25.6%), this does not seem to be a reliable biomarker in BTC for either durvalumab or pembrolizumab [5,6]. Thus, based on the TOPAZ-1 and KEYNOTE-966 clinical trials, there are now data that can be used to incorporate checkpoint inhibitors to first-line chemotherapy for all patients with BTC without a prior biomarker selection [5,6]. It is, however, a hugely unmet need to further explore potential biomarkers, since it is clear from the clinical trials performed up to now that not all patients benefit the same from these novel therapeutic approaches.

Our review highlights some of the main issues we are facing with the development of biomarkers for immunotherapy in BTC.

The clinical relevance of PD-L1 expression, MSI/MMRd, and TMB-H has been more widely investigated in other cancers [27].

In advanced non-small cell lung cancer (NSCLC) patients harboring a PD-L1 tumor proportion score (TPS) of ≥50%, immunotherapy alone has shown a clear benefit over chemotherapy doublets in first-line settings [28]. In BTC, a TPS > 50% is infrequent and sporadic data are precisely focused on this subgroup of patients [29]. TMB and MSI/MMRd are probably the most reliable biomarkers of immunotherapy efficacy, as testified by its consistency across different histologies [30,31]. Pembrolizumab has recently received the FDA’s full approval as agnostic treatment in patients with MSI-H/dMMR tumors based on results from the Phase 2 KEYNOTE-158, KEYNOTE-164, and KEYNOTE-051 trials. Few data are available on the efficacy of immunotherapy in BTC patients with TMB-H or MSI. Lemery et al. reported a 27% (3/11) response rate in MSI-H BTC patients treated with pembrolizumab [32]. Marabelle et al. reported a 40.9% ORR for MSI-H/dMMR cholangiocarcinoma patients included in the KEYNOTE-158 trial [33]. Moving to TMB, we removed 4.6% of TMB-H patients from our analysis. A new more recent pan-tumor analysis reported a 8.2% rate of TMB-H tumors (cut-off mut/Mb > 10) among the 21 cholangiocarcinomas included [34]. However, it should be noted that different cut-offs were used in the studies included in the present meta-analysis. Data on the efficacy of immunotherapy, specifically in TMB-H BTC patients, are limited, and no BTC patients with TMB-H were included in the KEYNOTE-158 trial [35,36]. Given the encouraging pan-tumor data, even in terms of tumor shrinkage, specific analysis on this subgroup of BTC patients is strongly required.

Immunotherapy administration alone (in the absence of MMR deficiency, high MSI, or high TMB) showed limited efficacy in patients with advanced BTC; however, we show in this study that these biomarkers are rarely present in BTC. The combination of durvalumab and pembrolizumab with standard first-line cisplatin and gemcitabine chemotherapy led to benefits in OS, and the combination of checkpoint inhibitor and chemotherapy is now considered a new standard of care in BTC. Despite PD-L1 being present in around one in four BTC patients, this is not a reliable predictive biomarker in BTC, and we have no way of identifying patients who may derive the most benefit.

On top of this, studies reporting novel biomarkers in BTC are lacking and the available studies are of poor quality. Moving ahead, learning what has already been developed in other diseases could be of much help. Available data from other malignancies in which immunotherapy has already gained a pivotal role in the treatment landscape, such as lung cancer, could serve as a guide. Moreover, new data on other biomarkers, such as INF-γ signature, tumor microenvironment, immune profile and spatial organization, and intratumoral microbiota, should be considered and prospectively evaluated in BTC patients who will receive immunoreactive therapies. Along these lines, the relevance of an immunoscore has already been recognized in other histologies [37], and an exhausted immune profile and low CD3/CD4 T cells in the tumor center are likely related to shorter survival in iCCA patients [38]. Despite searching for studies that explore these novel biomarkers in our systematic literature search, we failed to find significant studies. This highlights how much delayed we are in exploring these in BTC, while the field is much more advanced in other malignancies where immunotherapy has been the standard of care for much longer.

The challenge of combining multiple reports obtained with different technical procedures represents the key limitation of our investigation. We also acknowledge that we performed our analysis by only searching on Pubmed with the above-mentioned research strings, and some reports may not be screened. Overall, the quality of studies up to now is poor. Sample sizes are small, and most studies are retrospective. We need to make an effort to conceive and bring ahead prospective biomarker discovery studies together with drug development trials. These will permit a more accurate distinction between predictive/prognostic markers and impact assessment, which are otherwise challenging to obtain without randomization. Although our findings were similar across BTC subtypes, we probably need to explore further potential variabilities between different BTC groups, since it is now clear that they represent distinct entities with unique clinical and molecular characteristics.

Technical methods and standardizations for the assessment of immunotherapy-related biomarkers are pivotal points. For example, on PD-L1 expression, there is a highly inhomogeneous variety of retrospective literature studies on different antibodies, procedures, IHC platforms, positivity cut-off expression profiles on distinctive cells of the tumor microenvironment, kinds of expression (membranous versus cytoplasmic), and scoring systems. Although it has been suggested that a high PD-L1 expression is associated with response to immunotherapy in BTC, an optimal cut-off value has not been defined to date [39]. With regards to dMMR/MSI, the key is the concordance between IHC and PCR methods and the optimal detection flowchart. A recent ESMO recommendation document suggested that IHC can be used against four MMR proteins (MLH1, MSH2, MSH6, and PMS2) as part of the first assessment and PCR test in the case of indeterminate IHC results or a loss of only one heterodimeric subunit [40]. Furthermore, the relationship between this phenotype and other biomarkers, such as TMB or PD-L1, also deserves attention [40]. Turning to TMB, determining an optimal cut-off level for the definition of the TMB-H phenotype, with the best possible predictive value for immunotherapy efficacy, is the main point of debate. Based on the latest data, it seems like the cut-off of 10% is a useful tool in real life [22].

Finally, further exploring correlations between known and novel biomarkers could be something to explore further. As an example, for both TMB and MSI, correlation analysis with neoantigens’ tumor levels could be fascinating when considering the rationale for their value as biomarkers [19].

In conclusion, our study highlights the multiple unmet needs in the field of biomarkers for immunotherapy in BTC. Moving forward, we must plan and develop parallel translational studies that take place alongside drug development with adequate tissue and blood acquisition to develop novel biomarkers that move the field forward. This will require close collaboration between, clinical, translational, and basic researchers.

## Figures and Tables

**Figure 1 cells-12-02098-f001:**
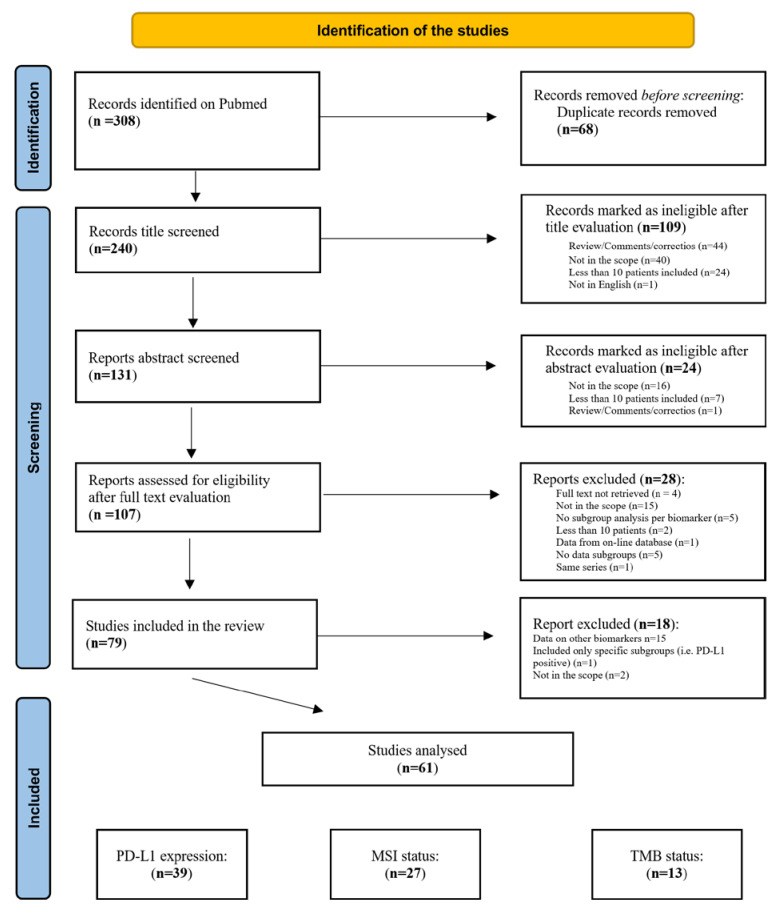
Study flowchart.

**Figure 2 cells-12-02098-f002:**
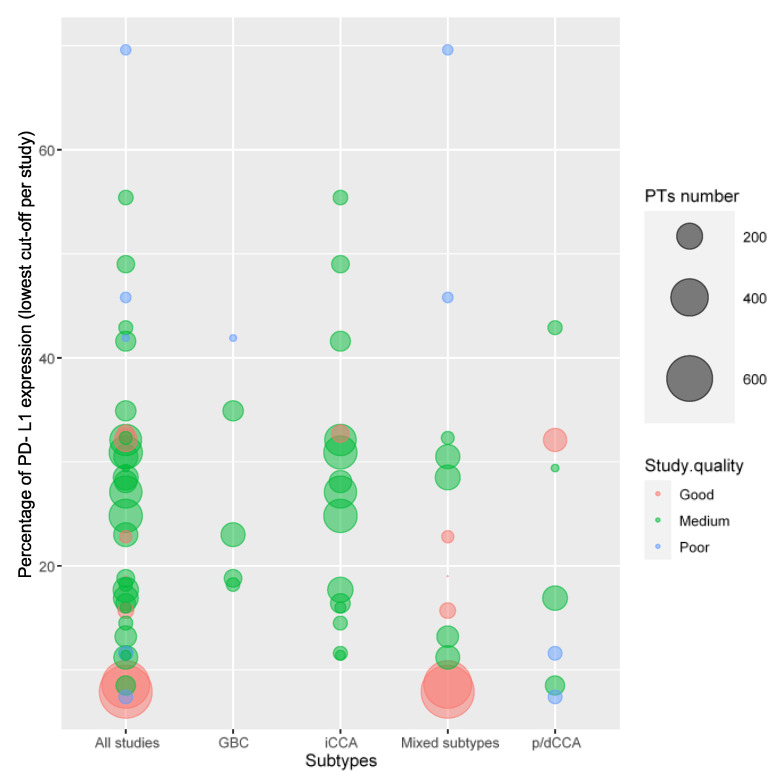
Percentage of PD-L1 expression in BTCs. iCCA, intrahepatic cholangiocarcinoma; p/dCCA, perihilar–distal cholangiocarcinoma; GBC, gallbladder cancer. “All studies” refers to all BTC studies combined in the same column. “Mixed subtypes” refers to studies that included more than one BTC subtype.

**Figure 3 cells-12-02098-f003:**
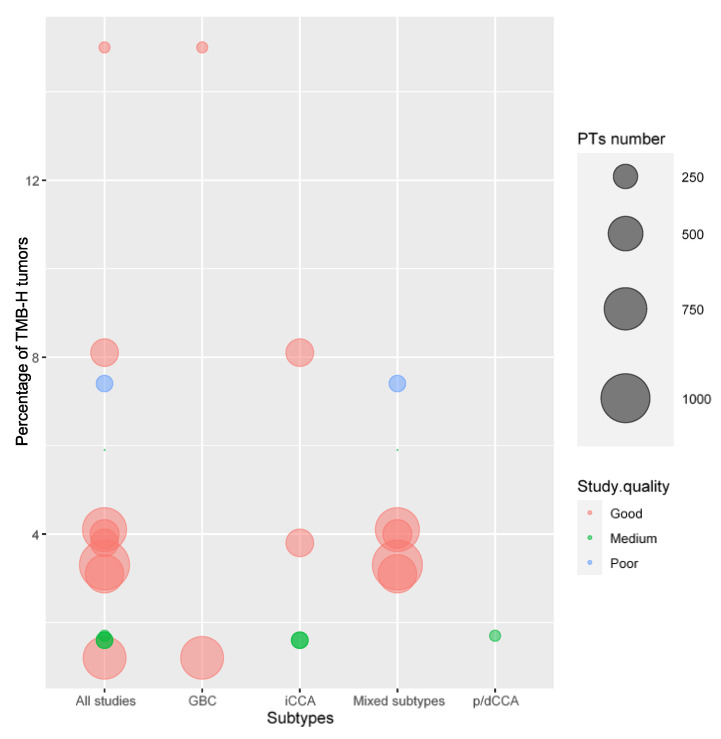
Percentage of TMB-H tumors in BTCs. iCCA, intrahepatic cholangiocarcinoma; p/dCCA, perihilar–distal cholangiocarcinoma; GBC, gallbladder cancer. “All studies” refers to all BTC studies combined in the same column. “Mixed subtypes” refers to studies that included more than one BTC subtype.

**Figure 4 cells-12-02098-f004:**
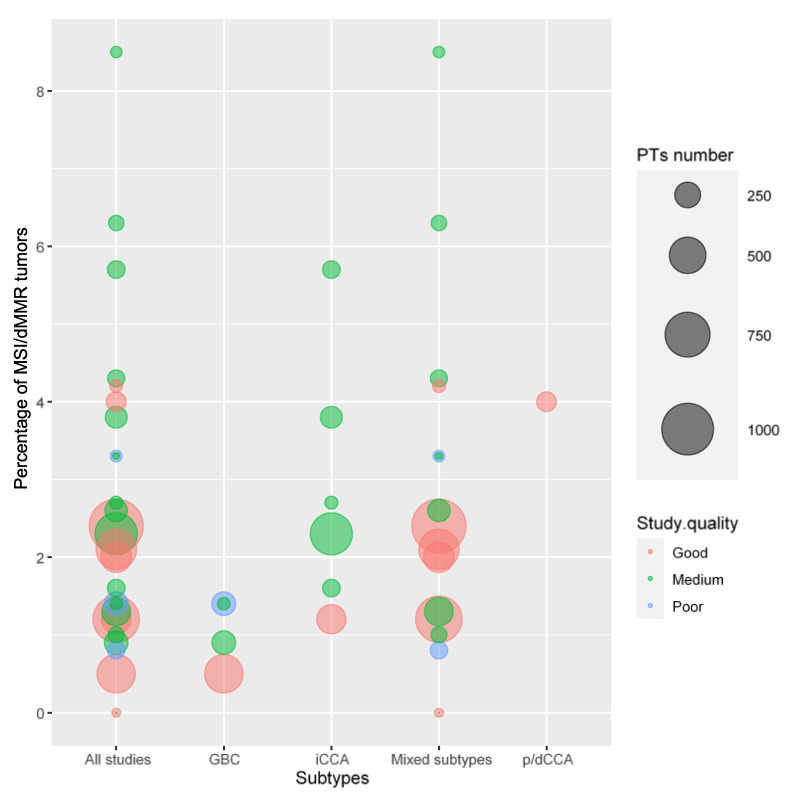
Percentage of MSI-H/dMMR in BTCs. iCCA, intrahepatic cholangiocarcinoma; p/dCCA, perihilar–distal cholangiocarcinoma; GBC, gallbladder cancer. “All studies” refers to all BTC studies combined in the same column. “Mixed subtypes” refers to studies that included more than one BTC subtype.

**Table 1 cells-12-02098-t001:** Risk-of-bias indicators.

Low-Risk-of-Bias Indicators	High-Risk-of-Bias Indicators
– prospective design – the consecutive selection of patients – multicentric study – central review of the biomarker – samples assessed by at least 2 blinded designated researchers – biomarker assessment based on non-standardized methodology – biomarker assessment based on a high-quality procedure – data available	– retrospective design (or not specified) – no-consecutive or possible bias in the selection of patients – monocentric study – samples assessed by only one non-blinded designated researcher – an assessment and review of the samples not described – no central review of the biomarker – biomarker assessment based on non-standardized or specified methodology – biomarker assessment based on poor-quality or non-specified procedures – no data available – discrepancy between patients included/evaluated

**Table 2 cells-12-02098-t002:** Characteristics of the included studies.

		All Studies (n = 61)	PD-L1 (n = 39)	TMB (n = 13)	MSI (n = 27)
**Study design**	Prospective trial	6 (9.8%)	2 (5.1%)	2 (15.4%)	3 (11.1%)
	Retrospective study	55 (90.2%)	37 (94.9%)	11 (84.6%)	24 (88.9%)
**Centers**	Multicenter	11 (18.0%)	6 (15.4%)	7 (53.8%)	9 (33.3%)
	Unicenter	50 (82.0%)	33 (84.6%)	6 (46.2%)	18 (66.6%)
**Reliability**	Good reliability	14 (23.0%)	7 (17.9%)	8 (61.5%)	9 (33.3%)
	Medium reliability	40 (65.6%)	27 (69.2%)	4 (30.8%)	15 (55.6%)
	Poor reliability	7 (11.5%)	5 (12.8%)	1 (7.7%)	3 (11.1%)
**Primary tumor**	iCCA only	21 (34.4%)	15 (38.4%)	5 (38.5%)	7 (25.9%)
	p/dCCA only	7 (11.5%)	7 (17.9%)	1 (7.7%)	1 (3.7%)
	GBC only	9 (14.8%)	5 (12.8%)	1 (7.7%)	3 (11.1%)
	Mixed subtypes	24 (39.3%)	12 (30.7%)	6 (46.2%)	16 (59.3%)

**Table 3 cells-12-02098-t003:** Description of patients of the studies included.

		All Cohorts (61 Studies)	PD-L1 (39 Studies)	TMB (13 Studies)	MSI (27 Studies)
**Total number of patients**		13,490	7778	6288	9127
**Total number of patients included for the analysis**		12,807	7682	6212	8470
**Mean number of total patients per study (95% CI)**		221 (151–292)	199 (117–282)	484 (217–751)	338 (194–482)
**Mean age (95% CI)**		62.7 (61.4–64.1)	62.5 (60.8–64.2)	61.1(60–62.1)	63.0 (61.4–64.6)
**Gender** **(mean)**	Data available	58/61 studies	31/39 studies	All studies	25/27 studies
	Male	115	105	229	169
	Female	103	96	254	167
**Stages**	Data available	2568 (19%)	1162 (14.9%)	947 (15.2%)	1152 (12.6%)
	I–II	1289 (50.2%)	769 (66.2)	358 (37.8%)	362 (31.4%)
	III–IV	1279 (49.8%)	393 (33.8)	589 (62.2%)	790 (68.6%)
**Disease status**	Data available	5697 (42.2%)	3831 (49.2%)	1157 (18.4%)	2877 (31.5%)
	Resected	4131 (72.5%)	2682 8 (70%)	190 (16.4%)	1639 (57%)
	Advanced	1566 (27.5%)	1149 (30%)	967 (83.6%)	1238 (43%)

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
