# Peer review of "Lacking Immunotherapy Biomarkers for Biliary Tract Cancer: A Comprehensive Systematic Literature Review and Meta-Analysis"

_cells, 2023, doi:10.3390/cells12162098_

Round 1
Reviewer 1 Report
I read with great interest the study entitled “Expression of PD-1/PD-L1 and other immunotherapy biomarkers in biliary tract cancer: Systematic Literature Review and Meta-analysis” by Frega et al focusing on the prevalence of immune-related biomarkers on biliary tract cancers. The study has many strengths; the methodology pursued is robust, and the topic at hand, while large, is nicely explored. However, I do have some concerns:
1. Why was only Pubmed searched for relevant articles? The authors should consider searching other databases (such as Embase) as well, in order to ensure that no potentially eligible studies are inadvertently omitted.
2. Overall, the methods and results of the study are well-articulated in the relevant sections of the manuscript. I am, however, confused as to whether patients with histologies other than iCCA, p/d CCA, GBC and AVC were included. The authors suggest that other histologic types were excluded, but report (in the results section) that “923 patients with other or unspecified histology” were also included. If that is the case, then why not consider excluding them as well?
3. Supplementary Table 1 exceeds the boundaries of the page and cannot be fully assessed.
4. There seems to be a technical error with Figures 2 and 3 as they appear blank.
5. I also suggest that the investigators provide a metric for interstudy heterogeneity (such as the Higgins I2) as is customary in meta-analyses.
6. The discussion, overall, is a weak point of the manuscript. There are multiple nonsensical sentences (i.e. “This will also help with the differentiation between predictive and prognostic impact of the markers we are eploring, challenging to do outside the setting of a randomized setting” and “technical issues at the time of assessment of immunotherapy-related biomarkers is a tumor-agnostic issue”). I suggest that the authors undertake a revision of the langue in this subsection of their manuscript.
7. Following on from the previous comment, a complete restructuring of the discussion is necessary to better delineate the findings of the study and how they fit within the existing literature. The limitations of the study need to be more concisely presented, in a comprehensible manner. Comparing the estimated prevalence of PD-1/TMB/MSI biomarkers in biliary tract cancers versus other types of malignancies would likely make for a useful discussion point.
Grammar, syntax and the meaning of sentences should be made more clear.
Author Response
I read with great interest the study entitled “Expression of PD-1/PD-L1 and other immunotherapy biomarkers in biliary tract cancer: Systematic Literature Review and Meta-analysis” by Frega et al focusing on the prevalence of immune-related biomarkers on biliary tract cancers. The study has many strengths; the methodology pursued is robust, and the topic at hand, while large, is nicely explored.
Thanks for your time in reviewing our manuscript and suggestions.
However, I do have some concerns:
- Why was only Pubmed searched for relevant articles? The authors should consider searching other databases (such as Embase) as well, in order to ensure that no potentially eligible studies are inadvertently omitted.
Response 1: we performed a systematic search on Pubmed and conference proceedings. We further omitted to include abstract proceeding data due to few data available. We did not plan a screening on other databases, such as Embase. Unfortunately, we cannot go back to the extraction phase of the studies, but we further addressed this point as a limitation of our analysis in the discussion.
- Overall, the methods and results of the study are well-articulated in the relevant sections of the manuscript. I am, however, confused as to whether patients with histologies other than iCCA, p/d CCA, GBC and AVC were included. The authors suggest that other histologic types were excluded, but report (in the results section) that “923 patients with other or unspecified histology” were also included. If that is the case, then why not consider excluding them as well?
Response 2: some papers included patients we different Histologies (such as hepato-cholangiocarcinoma). We specified that “A total of 13,490 patients were included in the 61 considered studies, 6,714 patients had iCCA, 2,083 p/dCCA (including 448 Klatskin tumors and 779 distal p/dCCA), 2,506 GBC, 37 AVC carcinoma and 923 patients with other or unspecified histology. Data from 12,870 patients were considered (some patients were excluded mainly due to mixed histology such as hepatobiliary carcinoma o restricted molecular subgroups).” Data of patients with unspecified histologies were not considered when possible. No more accurate distinction was possible by considering published data.
- Supplementary Table 1 exceeds the boundaries of the page and cannot be fully assessed.
Response 3: we modified the manuscript file. We apologize. We don’t know why this happened (the same for the figures). Please find the data on the new version. If you still cannot visualize them, we can ask the Editor and send them to you in another format.
- There seems to be a technical error with Figures 2 and 3 as they appear blank.
Response 4: we modified the manuscript file. We apologize. We don’t know why this happened (the same for the figures). Please find the Figures on the new version. If you still cannot visualize them, we can ask the Editor and send them to you in another format.
- I also suggest that the investigators provide a metric for interstudy heterogeneity (such as the Higgins I2) as is customary in meta-analyses.
Response 5: We do not calculate the Higgins heterogeity since this test is exstremelly useful when the number of studies in the meta-analysis is small. In our analysisis the concept of heterogeneity is more linked to the great variance among the technical procedures used in each studies rater than a “statistical” heterogeneity linked to different outcome (due to the smal number of studies).
- The discussion, overall, is a weak point of the manuscript. There are multiple nonsensical sentences (i.e. “This will also help with the differentiation between predictive and prognostic impact of the markers we are eploring, challenging to do outside the setting of a randomized setting” and “technical issues at the time of assessment of immunotherapy-related biomarkers is a tumor-agnostic issue”). I suggest that the authors undertake a revision of the langue in this subsection of their manuscript.
Response 6: Thanks for your suggestion. We extensively edited this section of the manuscript. You can find the changes highlighted in the text. “This will permit a more accurate distinction between predictive/prognostic markers and impact assessment, otherwise challenging to obtain without randomization.” Technical methods and standardization for the assessment of immunotherapy-related biomarkers are pivotal points.
- Following on from the previous comment, a complete restructuring of the discussion is necessary to better delineate the findings of the study and how they fit within the existing literature. The limitations of the study need to be more concisely presented, in a comprehensible manner. Comparing the estimated prevalence of PD-1/TMB/MSI biomarkers in biliary tract cancers versus other types of malignancies would likely make for a useful discussion point.
Response 7: Thanks for your suggestion. We added new parts highlighting what you suggested and following a new precise outline: results, comparison with other malignancies, what could be done, limitations of our analysis, and conclusion.
Reviewer 2 Report
This retrospective study is quite interesting and timely considering the new immunotherapy options arising for BTCs. The main conclusions shed light onto the difficulties in biomarkers and potential treatment options for BTCs, and that proper studies may help alleviate these difficulties. While the findings sound interesting it was difficult to interpret as Figures 1-4 seemed to be missing details for me? I am not sure if it is the version uploaded, an issue with the editorial system, or my computer - I did download several times and use several software programs to open the PDF so I am not sure. If some minor issues below could be addressed, as well as providing proper figures would be appreciated.
- There were some issues with the Figures in the PDF I was able to download. For some reason, it appears as if Figure 1 should have arrows or something but these are missing. Additionally, Figure 2, 3 and 4 should have some colors, etc. but it is just grey boxes. This is making it hard to assess the data. I am not sure if it is an issue with the PDF uploaded or the editorial site.
- There is a comment box and areas highlighted green that may indicate changes in track. These should be addressed and removed.
- In the Discussion "ariabilities" is probably meant to be variabilities.
- In the Discussion, this sentence seems to have a typo "this is not a reliable predictive biomarker reliable biomarker in BTC"
- The title is too vague. The title should be changed to better highlight the findings of the manuscript which, based on the Discussion, seem to be that current immunotherapy biomarkers are not sufficient for BTCs.
- The abstract does not accurately represent the conclusions. While it does discuss the data it does not reflect the main conclusions made by the authors, which is lack of a reliable biomarker and poor quality of studies thus far.
Author Response
Comments and Suggestions for Authors
This retrospective study is quite interesting and timely considering the new immunotherapy options arising for BTCs. The main conclusions shed light onto the difficulties in biomarkers and potential treatment options for BTCs, and that proper studies may help alleviate these difficulties. While the findings sound interesting it was difficult to interpret as Figures 1-4 seemed to be missing details for me? I am not sure if it is the version uploaded, an issue with the editorial system, or my computer - I did download several times and use several software programs to open the PDF so I am not sure.
If some minor issues below could be addressed, as well as providing proper figures would be appreciated.
Thanks for your opinion and suggestion. We uploaded a new modified version of our manuscript. If you cannot visualize the figures, we could send them to you in another format or file by asking the Editor.
- There were some issues with the Figures in the PDF I was able to download. For some reason, it appears as if Figure 1 should have arrows or something but these are missing. Additionally, Figure 2, 3 and 4 should have some colors, etc. but it is just grey boxes. This is making it hard to assess the data. I am not sure if it is an issue with the PDF uploaded or the editorial site.
We uploaded a new modified version of our manuscript. If you cannot visualize the figures, we could send them to you in another format or file by asking the Editor.
- There is a comment box and areas highlighted green that may indicate changes in track. These should be addressed and removed.
We removed all the boxes and previous comments/revisions.
- In the Discussion "ariabilities" is probably meant to be variabilities.
Thanks for your suggestion. It was variabilities. Please check the new version.
- In the Discussion, this sentence seems to have a typo "this is not a reliable predictive biomarker reliable biomarker in BTC"
Thanks for your suggestion. It was a typo. We modified the sentence in the new version.
- The title is too vague. The title should be changed to better highlight the findings of the manuscript which, based on the Discussion, seem to be that current immunotherapy biomarkers are not sufficient for BTCs.
We agree with your indication. Thanks. We modified the title accordingly: “Lacking Immunotherapy Biomarkers for Biliary Tract Cancer: A Comprehensive Systematic Literature Review and Meta-analysis”
- The abstract does not accurately represent the conclusions. While it does discuss the data it does not reflect the main conclusions made by the authors, which is lack of a reliable biomarker and poor quality of studies thus far.
Thanks again for your suggestion. We strongly agree and modified the abstract. In particular, we added this sentence: “We still lack a reliable biomarker, especially in patients with mismatch-proficient tumors, and must need to make an effort to conceive new prospective biomarker discovery studies.”
Round 2
Reviewer 1 Report
The authors appropriately addressed all concerns.
Reviewer 2 Report
The authors have addressed all of my previous concerns, and the figures are now able to be viewed.